# Propolis Alleviates Acute Lung Injury Induced by Heat-Inactivated Methicillin-Resistant *Staphylococcus aureus* via Regulating Inflammatory Mediators, Gut Microbiota and Serum Metabolites

**DOI:** 10.3390/nu16111598

**Published:** 2024-05-24

**Authors:** Zongze Li, Zhengxin Liu, Yuyang Guo, Shuangshuang Gao, Yujing Tang, Ting Li, Hongzhuan Xuan

**Affiliations:** School of Life Science, Liaocheng University, Liaocheng 252059, China; lizongze0204@126.com (Z.L.); zhengxinliu1999@163.com (Z.L.); guoyuyang1026@163.com (Y.G.); 15865232031@163.com (S.G.); 18864996286@163.com (Y.T.); 15269850505@163.com (T.L.)

**Keywords:** propolis, tree gum, heat-inactivated methicillin-resistant *Staphylococcus aureus*, acute lung injury, gut microbiota, serum metabolites

## Abstract

Propolis has potential anti-inflammatory properties, but little is known about its efficacy against inflammatory reactions caused by drug-resistant bacteria, and the difference in efficacy between propolis and tree gum is also unclear. Here, an in vivo study was performed to study the effects of ethanol extract from poplar propolis (EEP) and poplar tree gum (EEG) against heat-inactivated methicillin-resistant *Staphylococcus aureus* (MRSA)-induced acute lung injury (ALI) in mice. Pre-treatment with EEP and EEG (100 mg/kg, *p.o.*) resulted in significant protective effects on ALI in mice, and EEP exerted stronger activity to alleviate lung tissue lesions and ALI scores compared with that of EEG. Furthermore, EEP significantly suppressed the levels of pro-inflammatory mediators in the lung, including *TNF-α*, *IL-1β*, *IL-6*, and *IFN-γ*. Gut microbiota analysis revealed that both EEP and EEG could modulate the composition of the gut microbiota, enhance the abundance of beneficial microbiota and reduce the harmful ones, and partly restore the levels of short-chain fatty acids. EEP could modulate more serum metabolites and showed a more robust correlation between serum metabolites and gut microbiota. Overall, these results support the anti-inflammatory effects of propolis in the treatment of ALI, and the necessity of the quality control of propolis.

## 1. Introduction

Bacterial infections have become the second leading cause of death worldwide after ischemic heart disease [1]. A total of 7.7 million people worldwide die from bacterial infections each year, accounting for 13.6% of all deaths worldwide [2]. Five pathogens, including *Staphylococcus aureus*, *Escherichia coli*, *Streptococcus pneumoniae*, *Klebsiella pneumoniae* and *Pseudomonas aeruginosa*, accounted for 54.2% of the total deaths [3]. In addition, the abuse of antibiotics has led to the increasing resistance of bacteria, making the prevention and control of bacterial infections more severe [4]. It is estimated that 10 million people will die from drug-resistant infections by 2050 [5].

The Gram-positive superbug methicillin-resistant *Staphylococcus aureus* (MRSA) is resistant to almost all available antimicrobials, and can adapt to a wide range of host environments, posing a serious public health hazard [6]. Also, MRSA can cause acute respiratory distress syndrome (ARDS) and acute lung injury (ALI) [7]. At present, vancomycin is still the last line of defense against MRSA, but recently, strains resistant to vancomycin have been isolated clinically, making the prevention and control of MRSA increasingly urgent [8]. Natural products play an important role in anti-resistant bacteria and anti-inflammation due to their multi-component, multi-target and multi-pathway nature, and low side effects [9]. Therefore, searching for anti-MRSA and anti-inflammatory drugs from natural products and analyzing their mechanisms of action are crucial for preventing and treating MRSA infection.

Propolis is a sticky resinous substance that is gathered from the buds and bark of trees by western honey bees (*Apis mellifera* L.) [10]. The chemical composition of propolis is very complex, depending on the botanical and geographical origin of propolis [11]. Propolis from different parts of the world can be mainly divided into five types: *Populus type*, *Baccharis type*, *Clusia type*, *Maccaranga type* and *Mediterranean type*, and more than 600 components have been identified from these propolis [12]. Chinese propolis mainly belongs to “*populus*” propolis, and is rich in phenolic acids and flavonoids [13,14].

Propolis is known for its wide range of biological activities, such as antioxidant [15,16], anti-inflammatory [17], anticancer [18], immunoregulation [19,20], and antimicrobial effects [21]. Propolis has a remarkable antibacterial effect, which is mainly related to its main components, such as aromatic compounds and polyphenols [22]. Studies have tested over 600 bacterial strains and revealed that propolis effectively combats *Staphylococcus aureus*, *Escherichia coli*, *Streptococcus mutans*, *Streptococcus sobrinus*, *Actinomyces naeslundii*, *Pseudomonas aeruginosa*, *Bacillus subtilis*, *Listeria monocytogenes*, *Paenibacillus larvae* and *Micrococcus luteus*. A study confirmed that propolis is more effective against Gram-positive bacteria compared to Gram-negative ones [21]. Our previous studies have found that Chinese *Populus* propolis inhibits the proliferation of *S. aureus* and MRSA by inhibiting the formation of bacterial biofilms in vitro and having a synergistic effect with antibiotics [23]. Chinese red propolis can also inhibit the proliferation of *S. aureus* and MRSA by influencing cell metabolism [24]. 

Pathogenic bacteria invade the body and cause inflammation [25]. Normal inflammatory responses promote awareness of pathogenic microorganisms, but excessive inflammation leads to pathological damage, such as septic shock caused by acute cytokine storms and chronic persistent inflammation, which promotes the development of inflammation-related tumors and autoimmune diseases [26]. Studies have found that Turkey *Mediterranean* propolis has anti-inflammatory activities against mouse macrophage inflammation and acute lung injury induced by lipopolysaccharide (LPS) [27,28]. Chinese propolis can also inhibit the inflammatory damage of bovine mammary epithelial cells induced by heat-inactivated mastitis pathogens *E. coli* and *S. aureus* in vitro [29]. Brazilian propolis inhibits the LPS-induced mammary gland inflammatory damage in vivo [30]. More importantly, the lungs are characterized by a specialized milieu with individual microbial flora [31]. The lung microbiota undergo alterations in numerous respiratory disorders such as obstructive airway diseases, interstitial lung diseases, infections, and lung cancer [32]. The bidirectional gut–lung axis connecting the gut and pulmonary microbiota is widely acknowledged [33]. MRSA is a typical pathogen causing pneumonia [34]. Clinically, both MRSA and *Staphylococcus aureus* could cause pulmonary infections and an increase in inflammatory factors, including TNF-α, IL-1β, and IL-6 [35,36]. Moreover, MRSA could lead to a decrease in unclassified_f_Lachnospiraceae in the gut microbiota, as well as a reduction in the butyric acid in short-chain fatty acids [37]. However, the role of poplar propolis in the inflammation induced by drug-resistant bacteria is barely studied, and its effects on the inflammation induced by drug-resistant bacteria and its potential mechanism of action are still unclear.

In this study, we evaluated the effects of ethanol extracts from poplar propolis (EEP) and poplar gum (EEG) on heat-inactivated MRSA-induced acute lung injury (ALI) in mice via gut microbiota and serum metabolomics, and compared the differences in their activities.

## 2. Materials and Methods

### 2.1. Materials and Reagents

Propolis was collected from an apiary in Xintai county in Shandong province in 2022, and poplar was its plant source. EEG was gifted by a corporation in 2022, and the main plant sources were poplar buds. MRSA (ATCC 43300, China Center of Industrial Culture Collection, CICC, Beijing, China) was acquired from the Industrial Culture Collection of China. Anti-mouse F4/80 and anti-mouse NF-κB p65 were purchased from Wuhan Servicebio Technology Co., Ltd. (Wuhan, China). The RNA extraction kit was obtained from Beijing CarryHelix Biotechnology Co., Ltd. (Beijing, China), while the PrimeScript^TM^ RT Master Mix Reverse Transcription Kit and the TB Green^®^ Premix Ex Taq^TM^ II kit were purchased from TaKaRa (Dalian, China).

### 2.2. Sample Preparation and Chemical Composition Analysis

Frozen propolis was milled and then extracted with anhydrous ethanol, followed by ultrasonication at 50 °C for 3 h. After combining the supernatant from three separate extractions, the solution was subsequently filtered and concentrated under reduced pressure until reaching a consistent weight. The ethanol extracts from propolis (EEP) and EEG were stored at −20 °C. At the time of the experiment, EEP and EEG were redissolved in gum tragacanth, and the final concentration of ethanol in the gum tragacanth was less than 1% (*v*/*v*). MRSA was cultured according to previously reported methods, followed by heat inactivation at 80 °C for two hours, and then freeze-dried for subsequent experiments [24].

The chemical compositions of EEP and EEG were analyzed using UPLC-ESI-QTRAP-MS/MS. Samples were removed from the −80 °C conditions, and pre-cooled with 70% methanol (600 μL of extractant per 50 mg of sample) in order to vortex for 15 min. After that, the samples were centrifuged at 12,000 rpm for 3 min at 4 °C, and the supernatant was filtered with a microporous filter membrane (0.22 μm pore size) and stored in the injection vials for LC-MS/MS detection. The data acquisition instrumentation system mainly included Ultra Performance Liquid Chromatography (UPLC) (ExionLC^TM^ AD, https://sciex.com.cn/, accessed on 20 December 2022) and Tandem Mass Spectrometry (MS/MS) (Applied Biosystems 6500 QTRAP, https://sciex.com.cn/, accessed on 20 December 2022). Orthogonal Partial Least Squares Discriminant Analysis (OPLS-DA) was used to screen for differential metabolites. Screening criteria were variable importance in projection (VIP) ≥ 1, fold change (FC) ≥ 0.8, or FC ≤ 0.33. Model adequacy was assessed based on Q^2^ > 0.5 (predictive power) and *p* < 0.05.

### 2.3. Animals and Drug Administration

Male ICR mice weighing 32 ± 2 g (6 weeks old, *n* = 50) were purchased from Jinan Pengyue Laboratory Animal Breeding Co., Ltd. (Jinan, China) ((SCXK (LU) 20190003). Mice were maintained at a temperature of 22 ± 2.0 °C and 55% ± 15% humidity in a 12 h light/dark cycle environment. All the experiments on the animals were carried out following a protocol approved by the Ethics Committee of Liaocheng University (Approval No. 2023103107).

The experimental flow is shown in Figure 1. The mice were randomly assigned to one of five groups after one week of acclimatization (*n* = 10/group): (1) normal group (NOR group), mice were orally administered with 0.5% gum tragacanth; (2) model group (MRSA group), mice were orally administered with 0.5% gum tragacanth; (3) EEP group, mice were orally administered with EEP at a dose of 100 mg/kg; (4) EEG group, mice were orally administered with EEG at a dose of 100 mg/kg; (5) positive control group, mice were orally administered with 2 mg/kg dexamethasone (DXMS). All mice were pretreated for 7 days, and 24 h after the last administration, all mice except the normal group were injected with heat-inactivated MRSA (20 mg/kg) intravenously. The dose of propolis was consistent with those in previous studies [17]. The non-lethal dose of MRSA was derived according to previous reports [38]. All mice were euthanized 6 h later, and whole blood, lung tissues, and cecal feces were immediately obtained for subsequent studies.

### 2.4. Lung Histopathologic Analysis 

The right lung tissue was fixed with 4% paraformaldehyde and then subjected to hematoxylin–eosin (H&E) staining. The lung injury was estimated on a scale of 0 to 4 according to the scoring system outlined in the Official American Thoracic Society Workshop Report [39]: 0 for no or very mild injury, 1 for mild injury, 2 for moderate injury, 3 for severe injury, and 4 for very severe injury.

### 2.5. Immunohistochemical (IHC) Staining

Paraffin-embedded tissues underwent a 2 h heat treatment at 65 °C, followed by antigen retrieval using citrate buffer (pH 6.0) at 121 °C for 2 min. Endogenous catalase was blocked with 3% H_2_O_2_. After rabbit serum blocking, primary goat anti-rabbit immunoglobulin G antibodies (F4/80 and NF-κB) were incubated overnight. Secondary antibodies, horseradish enzyme-labeled goat anti-rabbit immunoglobulin G antibodies, were incubated for 30 min at room temperature. After staining with hematoxylin and 3,3′-diaminopropylamine, the sections were visualized under a light microscope, and the positive area was quantified using Image J 1.8.0 software.

### 2.6. Quantitative Real-Time Polymerase Chain Reaction (RT-qPCR) Analysis

The extraction of total RNA from the lung tissues of mice was conducted in accordance with the instructions of the TRIeasy^TM^ Total RNA Extraction Reagent. The NanoDrop 100 Ultra-Micro Spectrophotometer (Thermo Fisher, Waltham, MA, USA) was utilized to evaluate both the concentration and purity of the RNA. After the extraction, reverse transcription was carried out using the PrimeScript^TM^ RT Master Mix kit, followed by qPCR using the TB Green^®^Premix EX TaqTM reagent. The primers used in the study were obtained from Sangon Biotech Co., Ltd. (Shanghai, China). The 2^−ΔΔCT^ method was employed to determine the relative mRNA expression levels. For a detailed list of primer sequences, please refer to Table 1.

### 2.7. Gut Microbiota Analysis

Total genome DNA from fecal samples was extracted using the E.Z.N.A.^®^ Soil DNA Kit (Omega Bio-Tek Inc., Norcross, GA, USA). The 338F (50-ACTCCTACGGGAGGCAGCA-30) and 806R (50-GGACTACHVGGGTWTCTAAT-30) primers were used to amplify the V3–V4 regions of the 16S rRNA genes. The AxyPrep DNA Gel Extraction Kit (Axygen Biosciences, Union City, CA, USA) was used for the recovery and purification of amplification products. Amplification products were sequenced on the MiSeq platform (Illumina, San Diego, CA, USA). Sequence data were analyzed using the QIIME (1.9.1) software package. The clustering of operational taxonomic units (OTUs) was performed with a 97% similarity cutoff using UPARSE (http://drive5.com/uparse/, accessed on 20 November 2023). Each 16S rRNA gene sequence was analyzed for taxonomy by the RDP Classifier (https://rdrr.io/bioc/rRDP/, accessed on 20 November 2023) against the SILVA 119 16S rRNA database using a confidence threshold of 70%. The data were analyzed on the online platform of the Majorbio Cloud Platform (https://www.majorbio.com/, accessed on 20 November 2023).

α-diversity analysis involving Sobs and Shannon indices and β-diversity analysis employing principal co-ordinates analysis (PCoA) and Partial Least Squares Discriminant Analysis (PLS-DA) were conducted. Differential gut microbiota were identified using LEfSe multilevel species difference discriminant analysis, with a linear discriminant analysis (LDA) threshold > 2 and *p* < 0.05.

### 2.8. Short-Chain Fatty Acid (SCFA) Detection

About 50–100 mg of cecal feces was taken from mice, and we analyzed the short-chain fatty acid (SCFA) content using gas chromatography–mass spectrometry (GC-MS). The procedure followed our previously validated method [40]. The sample was centrifuged at 12,000 rpm for 10 min after adding 1 mL of acetone, and the resulting supernatant was extracted and subjected to gas chromatography using DB-FFAP.

### 2.9. Untargeted Metabolomics Analysis

Untargeted metabolomics was used to test the serum metabolites. The instrument used was an Agilent 6545 UHPLC/Q-TOF-MS. In detail, 400 μL of pre-cooled methanol/acetonitrile solvent (1:1, *v*/*v*) were added to 100 μL of serum sample to vortex for 30 s, and the samples were then sonicated for 10 min at 4 °C in a water bath and incubated overnight at −80 °C. The supernatant was extracted using cryovacuum spinning and dried. It was then centrifuged at 12,000 rpm for 15 min at 4 °C, followed by evaporation to dryness under a low-temperature vacuum. This process was repeated multiple times. Subsequently, the supernatant was redissolved in a methanol–acetonitrile solvent mixture (1:1, *v*/*v*), sonicated for 10 min in a water bath at 4 °C, and left overnight at −80 °C. After drying at 12,000 rpm for 15 min at 4 °C and evaporation under low-temperature vacuum, the solution was centrifuged at 12,000 rpm for 15 min at 4 °C, and the supernatant was used for the assay. To ensure quality, 5 μL from each sample tube was combined into the QC.

OPLS-DA was employed to discern intergroup variations and identify distinctive metabolites, with screening criteria set at VIP ≥ 1, FC ≥ 0.8, or FC ≤ 0.33. Model adequacy was assessed based on Q^2^ > 0.5 and *p* < 0.05. The identified differential metabolites were then matched against the KEGG database for further analysis.

### 2.10. Statistical Analyses

All data were presented as mean ± standard deviation (SD). Statistical analysis for group differences was conducted using one-way ANOVA followed by Tukey’s Honestly Significant Difference. If the data were not normally distributed, we used the Kruskal–Wallis test. GraphPad Prism 8.0.2 was employed for data analysis. Significance was denoted as * *p* < 0.05, ** *p* < 0.01, and *** *p* < 0.001. Spearman correlation analyses were carried out using the Metware Cloud platform (https://cloud.metware.cn, accessed on 20 December 2022), and the correlation strength was represented by the correlation coefficient (r-value) along with the corresponding *p*-value.

## 3. Results

### 3.1. The Chemical Compositions of EEP and EEG

The chemical compositions of poplar propolis and poplar gum were analyzed by UPLC-ESI-QTRAP-MS/MS, and a total of 1635 metabolites were identified in EEP and 1545 were identified in EEG. The Q^2^ of OPLS-DA was >0.9 and *p* < 0.05, indicating that the model selection was appropriate (Figure 2A,B). A total of 1040 differential metabolites were screened, and 859 metabolites in EEP had higher relative levels than EEG (Figure 2C), while 181 metabolites in EEG had higher relative levels than EEP. Differential metabolites in EEP and EEG included 208 flavonoids, 159 phenolic acids, 162 amino acids and derivatives, 93 lipids, 73 organic acids, 68 nucleotides and derivatives, 63 alkaloids, 41 lignans and coumarins, 32 terpenoids, 16 quinones, 5 tannins, and 120 others (Figure 2D).

### 3.2. EEP and EEG Alleviated MRSA-Induced ALI

The heat-inactivated MRSA-induced ALI model was established to evaluate the effects of EEP and EEG. Heat-inactivated MRSA significantly increased lung injury via the thickening and enlargement of alveolar walls, increasing the infiltration of inflammatory cells, interstitial hemorrhage and edema in lung tissues (Figure 3A). EEP and DXMS pretreatment significantly alleviated pulmonary edema, inflammatory cell infiltration, and alveolar wall thickening induced by inactivated MRSA. Notably, there were distinct differences between the scores of the EEP and EEG groups, with the EEP group scoring lower (Figure 3B).

The IHC staining of lung tissues also indicated that the positive areas of F4/80, an antigen on the surface of mature mouse macrophages crucial for immune response, inflammation, and tissue homeostasis, decreased obviously after EEP and DXMS pretreatment compared with the MRSA group (Figure 3C). Furthermore, the expression of NF-κB exhibited a significant decrease after EEP pretreatment, and a notable difference was observed between the EEP group and the EEG group (Figure 3D).

### 3.3. EEP and EEG Reduced the Levels of Pro-Inflammatory Cytokines

To investigate the effects of EEP and EEG treatments on the production of inflammatory cytokines in lung tissues, pro-inflammatory cytokines including *TNF-α*, *IL-1β*, *IL-6*, and *IFN-γ* were measured by RT-qPCR (Figure 3E–H), and the results indicate that EEP significantly decreased the levels of *TNF-α*, *IL-1β*, *IL-6*, and *IFN-γ*, and EEG treatment only reduced the levels of *IL-1β* and *IL-6*. DXMS pretreatment reduced the expression levels of *IL-1β*, *IL-6* and *TNF-α*.

### 3.4. EEP and EEG Increased the Diversity of Gut Microbiota and Modulated Community Composition

In the α-diversity analysis, the Sobs index and Shannon index represented the abundance and diversity of the whole community. As shown in Figure 4A,B, EEP pretreatment restored microbial abundance and diversity in ALI mice. However, neither were statistically significant compared to EEG pretreatment. For further insights into group similarities and differences, β-diversity analysis was conducted. PCoA based on the Bray–Curtis algorithm showed no significant separation between groups (Figure 4C). Considering potential inter-individual differences, PLS-DA was employed. The results show a significant separation between the EEP group and MRSA group (Figure 4D). Particularly, the MRSA group did not display a significant separation from the EEG group.

To comprehensively investigate the impacts of EEP and EEG on gut microbiota in the ALI mice model, we conducted a detailed analysis of the composition of the gut microbiota at both the phylum and genus levels. As shown in Figure 4E, the predominant gut microbiota included p_Firmicutes, p_Patescibacteria, p_Bacteroidota, p_Actinobacteriota, p_Desulfobacterota, and p_Verrucomicrobiota. At the genus level, the MRSA group altered the relative abundances of *g_Lactobacillus*, *g_Staphylococcus*, *g_norank_f_Muribaculaceae*, *g_unclassified_f_Lachnospiraceae*, *g_norank_f_norank_o_Clostridia_UCG-014* and *g_Akkermansia* (Figure 4F). EEP pretreatments partially increased the abundance of *g_norank_f_Muribaculaceae*, *g_unclassified_f_Lachnospiraceae*, *g_norank_f_Muribaculaceae*, *g_norank_f_norank_o_Clostridia_UCG-014*, *g_Enterorhabdus* and *g_Akkermansia*, and decreased the relative abundance of *g_Lactobacillus* and *g_Staphylococcus*, compared to MRSA. Notably, EEG pretreatment also modulates the gut microbiota to some extent. This suggests that both EEP and EEG have the potential to remodel the gut microbiota in the ALI mice model. To pinpoint specific differences in how the gut microbiota were influenced by EEP- and EEG-induced changes, LEfSe multilevel discriminant analysis was performed.

As shown in Figure 4G, LDA thresholds ≥2 were considered differential in the gut microbiota. The differential gut microbiota affected by heat-inactivated MRSA were o_Rhizobiales, f_Carnobacteriaceae, and *g_Atopostipes*. EEP-induced alterations involved *g_GCA-900066575*, *g_Tyzzerella*, *g_Eubacterium_xylanophilum*, *g_unclassified_f_Oscillospiraceae*, o_Enterobacterales, *g_NK4A214_group* and *g_Ruminococcus*. EEG-induced changes encompassed f_Aerococcaceae, o_Bacillales, *g_Aerococcus*, f_Bacillaceae, *g_Pseudogracilibacillus* and c_Actinobacteria. In summary, both EEP and EEG have the capacity to modify the composition of gut microbiota in heat-inactivated MRSA-induced ALI mice. These alterations in the gut microbiota had the potential to further influence the immune response and metabolite changes in vivo, ultimately contributing to alleviating excessive inflammatory responses.

### 3.5. Effects of EEP and EEG on the SCFAs in Heat-Inactivated MRSA-Induced ALI Mice

SCFAs serve as crucial energy sources for intestinal cells and play pivotal roles in immunomodulation and lipid metabolism [41,42]. Our findings demonstrate that acetic acid, propionic acid, and butyric acid concentrations were notably reduced in the MRSA group. Particularly noteworthy are the significant increases in acetic acid and valeric acid following EEP pretreatment. Conversely, EEG pretreatment significantly raised acetic acid, butyric acid, and total SCFAs. However, pretreatments with EEP and EEG led to elevated levels of acetic acid, butyric acid, propionic acid, and total SCFAs (Figure 5A–E). These results indicate that both EEP and EEG interventions can partially restore SCFAs, with the most notable disparity observed in butyric acid between the two pretreatments.

### 3.6. Analysis of Serum Metabolites

To gain further insights into the distinct effects of EEP and EEG on metabolites and their mechanisms, we conducted an untargeted metabolomics analysis of serum samples. Three-dimensional principle component analysis (3D-PCA) revealed no significant separation among the NOR, MRSA, EEP, and EEG groups in either positive or negative ion modes (Figure 6A,B). Consequently, we employed the OPLS-DA model to screen and delineate group variances. The Q^2^ values of the model exceeded 0.5 with *p* < 0.05 in both ion modes, indicating that the model was appropriate (Appendix A). As shown in Figure 6C,D, the OPLS-DA score plot showed a clear separation between the groups. Volcano plot analysis unveiled 119, 98, and 89 differential metabolites in the NOR, EEP, and EEG groups, respectively, relative to the MRSA group (Figure 6E–G). Furthermore, we observed that EEP pretreatment modulated 25 differential metabolites in the MRSA group, with 21 differential metabolites showing restoration to normal levels, while EEG pretreatment influenced 15 differential metabolites in the MRSA group, with 12 differential metabolites exhibiting normalization (Figure 6H and Table 2). Notably, our analysis revealed six differential metabolites in both the EEP group and EEG group, predominantly comprising lipids and amino acids (Figure 7A–F).

To gain deeper insights into the distinct regulatory mechanisms of EEP and EEG, we conducted a comprehensive analysis by aligning the differential metabolites from each group with the KEGG database for metabolic pathway exploration. The KEGG enrichment classification revealed that the NOR group’s differential metabolites were associated with 19 pathways, with primary enrichments in Bile secretion, alpha-Linolenic acid metabolism, Glycosylphosphatidylinositol (GPI)-anchor biosynthesis, Glycerophospholipid metabolism, Retrograde endocannabinoid signaling, Amino sugar and nucleotide sugar metabolism, and Cysteine and methionine metabolism (Figure 7G). Notably, PE substances exhibited the highest match with diverse metabolic pathways. In the EEP group, the differential metabolites were aligned with 39 metabolic pathways, showcasing major enrichments in Bile secretion, Glycerophospholipid metabolism, Glycosylphosphatidylinositol (GPI)-anchor biosynthesis, Retrograde endocannabinoid signaling, Pentose and glucuronate interconversions, Ascorbate and aldarate metabolism, Phenylalanine metabolism, and Purine metabolism (Figure 7H). Remarkably, PE analogs, LysoPC analogs, and amino acid analogs emerged as the predominant contributors to these pathways. Differential metabolites in the EEG group were linked to 20 metabolic pathways, primarily enriched in Glycosylphosphatidylinositol (GPI)-anchor biosynthesis, Glycerophospholipid metabolism, and Retrograde endocannabinoid signaling pathways (Figure 7I). Notably, PE substances were associated with a higher number of metabolic pathways. These findings suggest that the EEP group exhibited a broader range of matched metabolic pathways, while the EEG group and MRSA group shared predominantly similar pathways. Furthermore, lipid metabolism and amino acid metabolic pathways were identified as potential areas of metabolic significance.

### 3.7. Correlation Analysis between Gut Microbiota and Serum Metabolites

Spearman correlation analysis was used to explore the potential relationships between gut microbiota and serum metabolites. As shown in Figure 8, the analysis revealed that the differential gut microbiota in the EEP group exhibited a negative correlation with metabolites. For example, *g_Tyzzerella* and *g_Eubacterium_xylanophilum_group* in the propolis pretreatment group were negatively correlated with PE (18:4(6Z,9Z,12Z,15Z)/24:1(15Z)) and PE (20:4(5Z,8Z,11Z,14Z)/P-18:1(11Z)). N-(1-Deoxy-1-fructosyl)valine, 2-Ethylacrylylcarnitine and 5′-Methylthioadenosine were negatively correlated with *g__GCA-900066575*. PE(20:4(5Z, 8Z,11Z,14Z)/P-18:1(11Z)) and N-(1-Deoxy-1-fructosyl)valine were also negatively correlated with *g_NK4A214_group*. *g_Ruminococcus* was negatively correlated with 5′-Methylthioadenosine, N-(1-Deoxy-1-fructosyl)valine and L -γ-Glutamyl-β-phenyl-β-L-alanine. These results suggest that our propolis pretreatment may further influence metabolite changes in vivo through the gut microbiota. However, there was no significant correlation observed between differential microbiota and metabolites in the EEG group and MRSA group. These findings imply that our EEP pretreatment group might exert an influence on in vivo metabolite changes through the modulation of gut microbiota.

## 4. Discussion

In this study, we investigated for the first time the effects of propolis and tree gum on heat-inactivated MRSA-induced acute lung injury and the difference in efficacy by gut microbiota and serum metabolomics. Both propolis and tree gum were able to alleviate MRSA-induced acute lung injury, but the efficacy of propolis was significantly better than that of tree gum, and the effects of propolis on the gut microbiota of ALI in mice were more significant, with stronger correlations with serum metabolites. These results suggest that the difference in chemical components between propolis and tree gum determines the difference in their efficacy, and the results of the present study provide a basis for the treatment of ALI with propolis while indicating the necessity for the quality control of propolis.

The main manifestations of ALI include lung tissue lesions, macrophage proliferation, increased inflammatory factors, and the activation of related signaling pathways [43]. Mussbacher et al. found that macrophages play a key role in immune and inflammatory regulation, and the activation of NF-κB is a critical step during the process [44]. In addition, Búfalo MC et al. found that propolis and its component caffeic acid inhibited macrophage inflammatory responses [45]. Our results suggest that propolis significantly reduced lung histopathology, the release of inflammatory factors, macrophage proliferation, and NF-κB expression in ALI in mice compared with tree gum.

Several studies have shown that the gut microbiota of patients with severe pneumonia undergo dysbiosis, with a decrease in the number of beneficial genera and an increase in the number of pathogenic genera, and an imbalance in the intestinal microecology [46,47,48]. Moreover, the application of antibiotics leads to more dysbiosis of the gut microbiota, decreasing intestinal resistance, increasing the infiltration of inflammatory cells and lymphocytes in the lungs, and exacerbating lung damage [49]. Studies have found significant differences between the gut microbiota of patients with severe pneumonia and those of healthy people, while the proportion of pathogenic bacteria such as *Staphylococcus* spp. increased significantly in patients with severe pneumonia, and the proportions of beneficial commensal bacteria such as *Bifidobacterium*, *Ruminococcus*, *Muribaculaceae*, and *Enterorhabdus* were significantly lower than in the healthy population [46]. The imbalance of the gut microbiota could overactivate patients’ systemic inflammatory stress, promoting the progression of severe pneumonia. Propolis pretreatment significantly reduced the elevation of *g_Staphylococcus* caused by MRSA, and *g_Ruminococcus*, *g_norank_f_Muribaculaceae* and *g_Enterorhabdus* were also significantly upregulated in the propolis pretreated group, and *Ruminococcus* and *Enterobacterales* were the dominant bacteria in the propolis group.

Notably, ALI in mice induced by the heat inactivation of MRSA increased the abundance of *g_Lactobacillus*. However, propolis pretreatment decreases its abundance. Although *g_Lactobacillus* is representative of beneficial bacteria, its excessive abundance tends to disrupt the homeostasis of gut microbes in vivo, which in turn affects the production of SCFAs [50]. After pretreatment with propolis, we also found that *g_GCA-900066575*, *g_Tyzzerella* and *g_Eubacterium_xylanophilum* belonging to f_Lachnospiraceae were the dominant mcrobiota. f_Lachnospiraceae belong to the Firmicutes, one of the more important families that produce SCFAs, and some of its families are involved in the metabolism of bile acids [51]. *g_unclassified_f_Oscillospiraceae* and *g_NK4A214_group* are also dominant bacteria in ALI in mice after propolis pretreatment, which all belong to the o_Oscillospirales. o_Oscillospirales have a strong connection with human health, can produce SCFAs, and may be representative of the next generation of beneficial bacteria [52]. In the tree gum pretreatment group, f_Bacillaceae and *g_Pseudogracilibacillus* belonging to o_Bacillales were the dominant microbiota. o_Bacillales are one of the more important constituents of the Firmicutes, which play an important role in the regulation of intestinal microbial homeostasis and the immune response of the host [53].

Changes in the composition and structure of the gut microbiota not only affect human health, but also the production of SCFAs [41]. SCFAs are metabolites produced by intestinal microorganisms fermenting dietary fibers, and are one of the major sources of energy for the body and the colonic epithelium [42]. SCFAs also play an important role in the immune response of the host and in metabolism. Numerous studies have shown that the gut microbiota can modulate SCFAs to alleviate acute lung injury [54,55,56]. In the present study, heat-inactivated MRSA induced a decrease in SCFAs in ALI in mice, and propolis pretreatment elevated the concentrations of acetic acid and valeric acid to reverse these changes. Differently, tree gum pretreatment could elevate the concentrations of acetic acid and butyric acid to alleviate heat-inactivated MRSA-induced ALI in mice. It is noteworthy that the dominant bacteria in the propolis pretreatment group, f_Lachnospiraceae and o_Oscillospirales, were the major gut microbiota that produced SCFAs. These results suggested that propolis pretreatment has a close link to the production of SCFAs, mediated by the gut microbiota.

Changes in the gut microbiota are also closely linked to host metabolism. The analysis of serum metabolomics can help us understand the mechanisms of disease occurrence, as well as obtain corresponding biomarkers [57]. Zhou et al. found that lung lipids play a crucial role in maintaining the structural integrity and function of lung tissue [58]. Lipids, as cell membrane components and energy sources, are substances that play an important role in immune regulation and have signaling transduction functions [59]. In ALI, lung lipid homeostasis may be severely disrupted, affecting the normal function of lung tissue [60]. In addition, ALI is associated with the dysregulation of amino acid metabolism [61]. Amino acids have important regulatory roles in the developmental process of lung injury, such as affecting the body’s immunity, regulating the release of inflammatory factors, and influencing oxidative stress [33]. In the present study, propolis pretreatment was able to affect lipid and amino acid groups more in vivo compared to tree gum pretreatment.

Importantly, the metabolic pathways affected by propolis pretreatment include Glycerophospholipid metabolism, Glycosylphosphatidylinositol (GPI)-anchor biosynthesis, Retrograde endocannabinoid signaling and Phenylalanine metabolism. Notably, PE (18:4(6Z,9Z,12Z,15Z)/24:1(15Z)), PE (14:1(9Z)/14:1(9Z)), and PE (20:4(5Z,8Z,11Z,14Z)/P-18:1(11Z)) are the major components of glycerophospholipid metabolism. Propolis pretreatment restored the levels of PE (18:4(6Z,9Z,12Z,15Z)/24:1(15Z)) and PE (20:4(5Z,8Z,11Z,14Z)/P-18:1(11Z)) to normal. These substances play an important role in the regulation of lipid metabolism mechanisms.

Furthermore, we found that beneficial bacteria in the propolis pretreatment group showed a strong negative correlation with lipid and amino acid metabolites in the correlation analysis, which suggests that an increase in the abundance of beneficial bacteria may restore the metabolites to normal levels, also demonstrating that the gut microbiota affected the metabolic processes of the host.

## 5. Conclusions

In summary, we first evaluated the effects of propolis on ALI induced by drug-resistant bacterial infection, and the results demonstrate that both propolis and tree gum could alleviate ALI induced by heat-inactivated MRSA by reducing inflammatory cytokines, and modulating gut microbiota, SCFAs, and serum metabolites. The effects of propolis on ALI were more significant than those of tree gum, indicating that the differences in chemical compositions between propolis and tree gum determined their different anti-inflammatory effects. This study provides evidence of the utility of propolis in the treatment of drug-resistant bacterial infections, and suggests the necessity of propolis quality control.

## Figures and Tables

**Figure 1 nutrients-16-01598-f001:**
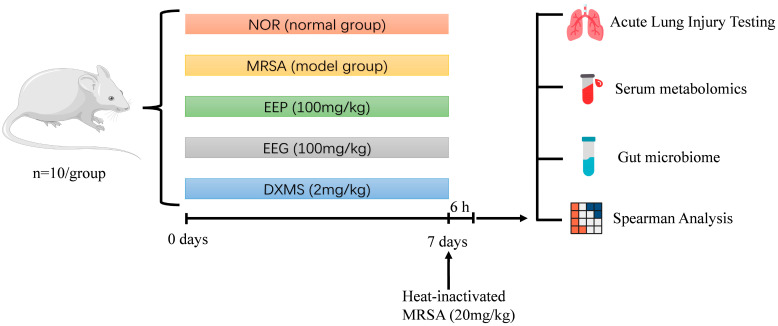
The workflow of the animal experiments.

**Figure 2 nutrients-16-01598-f002:**
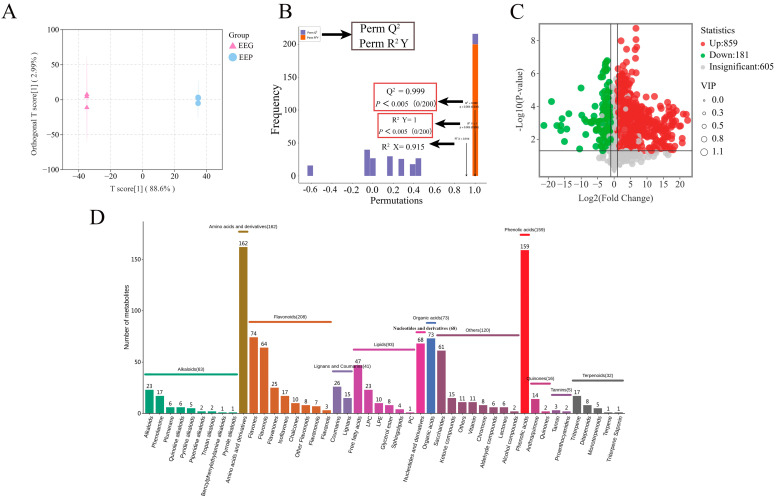
The chemical compositions of EEP and EEG. (**A**) OPLS-DA score plots. (**B**) OPLS-DA validation plot. (**C**) Volcano plot of differential metabolites. (**D**) Classification of differential metabolites.

**Figure 3 nutrients-16-01598-f003:**
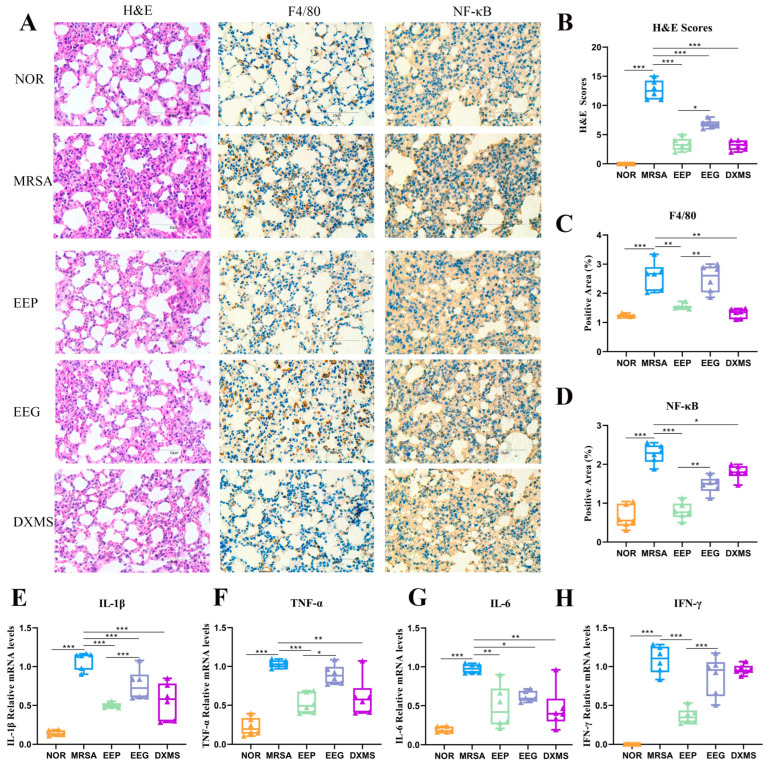
Effects of EEP and EEG on heat-inactivated MRSA-induced ALI in mice. (**A**) H&E and IHC staining (magnification 400×, *n* = 6, scale bar = 10 μm). H&E staining was used to observe lung histopathology, and IHC staining was used to observe F4/80 and NF-κB expression. (**B**) Lung injury score based on H&E staining. (**C**) Percentage of F4/80 positive areas. (**D**) Percentage of NF-κB-positive areas. (**E**–**H**) The mRNA expression levels of IL-1β, TNF-α, IL-6 and IFN-γ, respectively. Data are shown as means ± SD. * *p* < 0.05, ** *p* < 0.01, and *** *p* < 0.001 versus the MRSA group, *n* = 6.

**Figure 4 nutrients-16-01598-f004:**
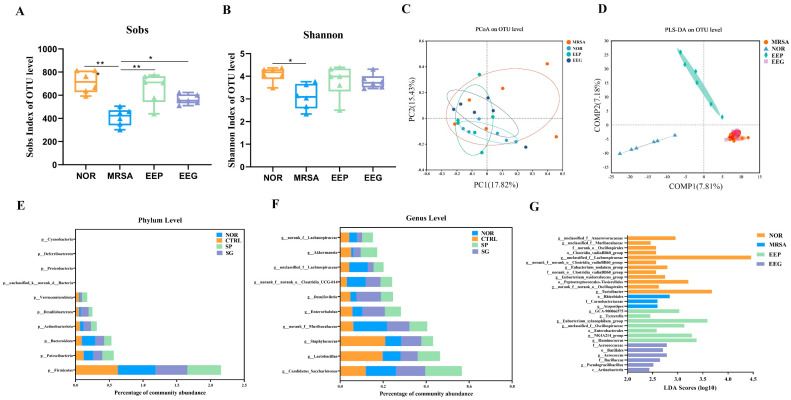
EEP and EEG analysis of gut microbiota in heat-inactivated MRSA-induced ALI mice. (**A**,**B**) α-diversity analysis. Sobs and Shannon indices are at the OUT level. Data are shown as means ± SD. * *p* < 0.05, ** *p* < 0.01 versus the MRSA group. (**C**,**D**) β-diversity analysis. PCoA and PLS-DA analyses at the OUT level. Subgroup tests were performed by Adonis. (R^2^ = 0.16, *p* < 0.05). (**E**,**F**) Gut microbiota composition at the phylum level and genus level. (**G**) The NOR group, MRSA group, EEP group, and EEG group were analyzed using LEfSe analysis from phylum to genus level. Data were analyzed using the one-against-all multi-group comparison strategy, with LDA thresholds ≥2.

**Figure 5 nutrients-16-01598-f005:**
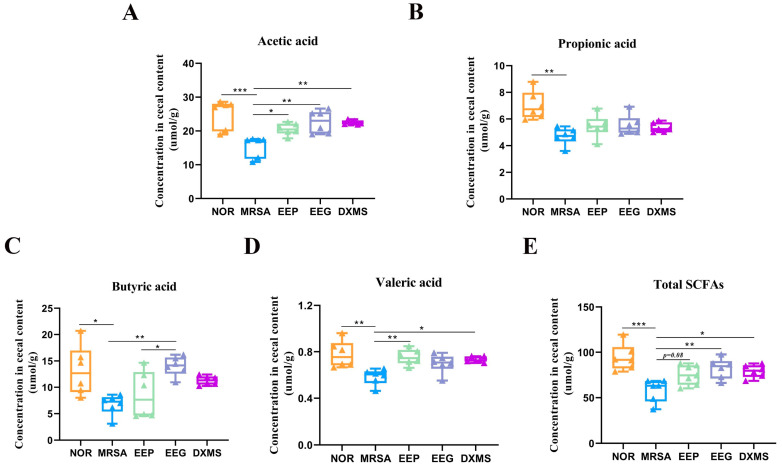
Effect of EEP and EEG on SCFAs in heat-inactivated MRSA-induced ALI mice. (**A**–**E**) Concentrations of acetic acid, propionic acid, butyric acid, valeric acid, and total SCFAs in feces from the cecum of mice. Data are shown as means ± SD. * *p* < 0.05, ** *p* < 0.01, and *** *p* < 0.001 versus the MRSA group, *n* = 6.

**Figure 6 nutrients-16-01598-f006:**
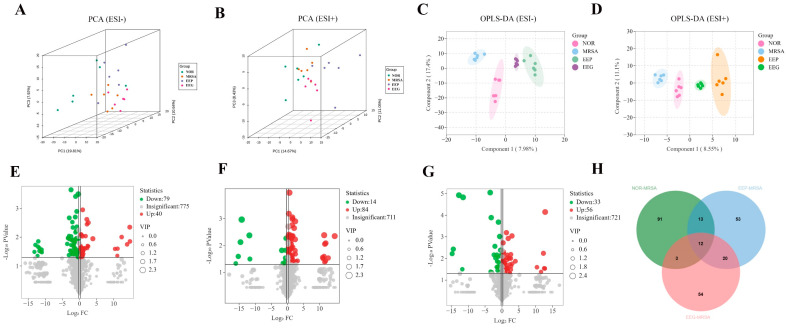
Serum metabolomic analysis of heat-inactivated MRSA-induced ALI mice by EEP and EEG. (**A**,**B**) 3D-PCA analysis in negative and positive ion modes. (**C**,**D**) OPLS-DA score plots. The predictive parameters of the model in negative ion mode are Q^2^ = 0.699, *p* < 0.05. The predictive parameters of the model in positive ion mode are Q^2^ = 0.552, *p* < 0.05. (**E**–**G**) Volcano plots show the screening of differential metabolites of the NOR group, the EEP group, and the EEG group versus the MRSA group in both positive and negative ion modes. (**H**) Venn diagram of differential metabolites.

**Figure 7 nutrients-16-01598-f007:**
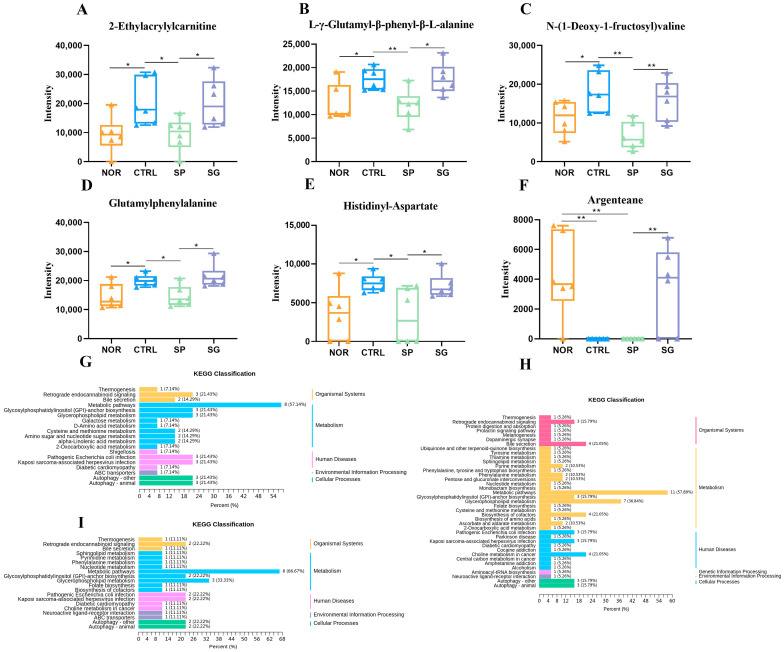
Differential metabolites of EEP and EEG, and KEGG pathway analysis. (**A**–**F**) The corresponding peak intensities of the six substances. Data are shown as means ± SD. * *p* < 0.05, ** *p* < 0.01 versus the MRSA group, *n* = 6. (**G**–**I**) The pathways matched by differential metabolites in the EEP, EEG, and MRSA groups compared to the NOR group, with the vertical coordinates representing the various types of pathways and the horizontal coordinates representing the percentage of pathways.

**Figure 8 nutrients-16-01598-f008:**
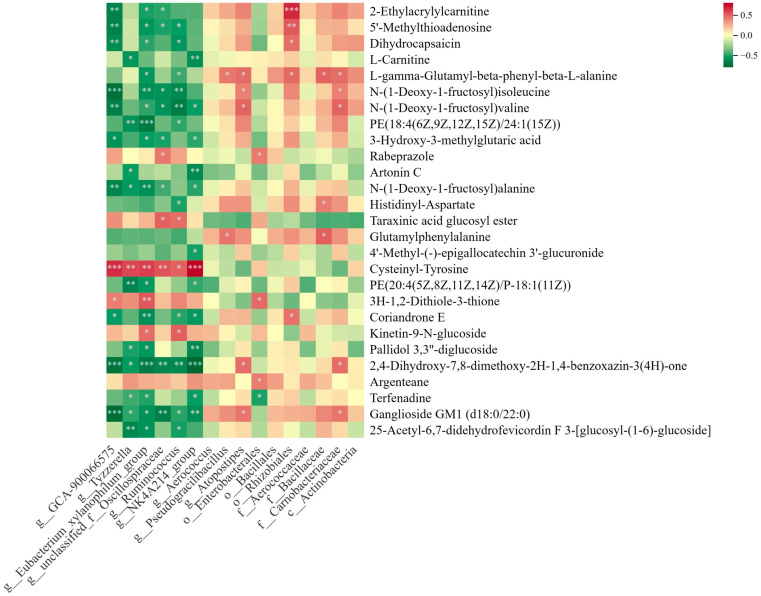
Spearman correlation analysis. Horizontal coordinates represent differential gut microbiota and vertical coordinates represent differential metabolites. Differences are indicated by asterisks (* *p* < 0.05, ** *p* < 0.01, and *** *p* < 0.001). The red color represents a positive correlation, and a green color represents a negative correlation. The shift in color from dark to light represents the strength of the correlation.

**Table 1 nutrients-16-01598-t001:** Primers for RT-qPCR.

Gene	Primer Sequences (5′-3′)
* TNF-α *	Forward: CCACGCTCTTCTGTCTACTG Reverse: CCACGCTCTTCTGTCTACTG
* IL-6 *	Forward: CTCTGCAAGAGACTTCCATCC Reverse: GAATTGCCATTGCACAACTC
* IL-1β *	Forward: CCAACAAGTGATATTCTCCATGAG Reverse: ACTCTGCAGACTCAAACTCCA
* IFN-γ *	Forward: GACTGTGATTGCGGGGTTGT Reverse: GGCCCGGAGTGTAGACATCT

**Table 2 nutrients-16-01598-t002:** Differential metabolites affected after EEP and EEG pretreatments.

No	Ion Mode	Compound	NOR vs. MRSA				EEP vs. MRSA				EEG vs. MRSA			
			VIP	*p*-Value	FC	Type	VIP	*p*-Value	FC	Type	VIP	*p*-Value	FC	Type
1	-	Rabeprazole	1.48	0.03	0.29	Down	1.59	0.04	1.93	Up	-	-	-	-
2	-	3-Hydroxy-3-methylglutaric acid	1.87	0.001	1.88	Up	1.66	0.02	1.80	Up	-	-	-	-
3	-	N-(1-Deoxy-1-fructosyl)alanine	1.42	0.03	1.79	Up	1.80	0.01	2.40	Up	-	-	-	-
4		Histidinyl-Aspartate	1.41	0.02	2.15	Up	1.76	0.02	2.35	Up	-	-	-	-
5	-	Glutamylphenylalanine	1.44	0.02	1.38	Up	1.87	0.01	1.37	Up	-	-	-	-
6	-	Coriandrone E	1.60	0.007	1.34	Up	1.58	0.03	1.40	Up	-	-	-	-
7	-	Ganglioside GM1 (d18:0/22:0)	1.39	0.03	2.27	Up	1.55	0.04	1.88	Up	-	-	-	-
8	+	2-Ethylacrylylcarnitine	1.67	0.02	2.18	Up	1.61	0.02	2.15	Up	-	-	-	-
9	+	5′-Methylthioadenosine	1.99	0.009	1.79	Up	1.53	0.03	1.61	Up	-	-	-	-
10	+	L-γ-Glutamyl-β-phenyl-β-L-alanine	1.78	0.02	1.41	Up	1.80	0.009	1.46	Up	-	-	-	-
11	+	N-(1-Deoxy-1-fructosyl)isoleucine	1.61	0.03	1.55	Up	2.07	0.001	2.89	Up	-	-	-	-
12	+	Retapamulin	1.49	0.03	0.60	Down	1.80	0.009	1.61	Up	-	-	-	-
13	+	N-(1-Deoxy-1-fructosyl)valine	1.56	0.03	1.57	Up	2.02	0.001	2.71	Up	1.60	0.02	0.80	Down
14	-	Cysteinyl-Tyrosine	1.64	0.01	0.71	Down	1.91	0.005	0.75	Down	1.56	0.02	0.62	Down
15	-	3H-1,2-Dithiole-3-thione	1.94	0.001	0.42	Down	1.74	0.01	0.64	Down	1.61	0.02	0.33	Down
16	-	Kinetin-9-N-glucoside	1.66	0.009	0.18	Down	1.54	0.04	0.37	Down	1.70	0.03	0.0002	Down
17	+	Argenteane	1.72	0.01	0.0002	Down	1.95	0.0	3188.3	Up	1.91	0.004	1.37	Up
18	-	Artonin C	1.54	0.01	1.26	Up	1.50	0.03	1.25	Up	2.14	0.001	7.79	Up
19	-	PE(20:4(5Z,8Z,11Z,14Z)/P-18:1(11Z))	1.72	0.003	4.52	Up	2.16	0.001	5.27	Up	2.37	0.001	7067.5	Up
20	-	Pallidol 3,3″-diglucoside	1.89	0.002	4.38	Up	1.96	0.005	5.28	Up	1.75	0.009	6.14	Up
21	-	2,4-Dihydroxy-7,8-dimethoxy-2H-1,4-benzoxazin-3(4H)-one	1.90	0.004	26,450.0	Up	2.11	0.004	26,450.0	Up	1.74	0.01	1.73	Up
22	-	Terfenadine	1.68	0.009	3.44	Up	1.68	0.02	1.67	Up	1.66	0.03	1.71	Up
23	+	Dihydrocapsaicin	1.57	0.04	1.79	Up	1.93	0.004	3.16	Up	1.71	0.01	1.30	Up
24	+	L-Carnitine	1.89	0.008	1.44	Up	1.69	0.01	1.24	Up	1.74	0.03	2.66	Up
25	+	PE(18:4(6Z,9Z,12Z,15Z)/24:1(15Z))	1.75	0.02	3.02	Up	1.69	0.02	3.10	Up	1.65	0.02	2.93	Up
26	-	Taraxinic acid glucosyl ester	1.80	0.006	0.51	Down	-	-	-	-	1.47	0.04	1.66	Up
27	-	4′-Methyl-(-)-epigallocatechin 3′-glucuronide	1.45	0.03	1.50	Up	-	-	-	-	1.71	0.02		Up
28	+	25-Acetyl-6,7-didehydrofevicordin F 3-[glucosyl-(1-6)-glucoside]	1.56	0.02	1314	Up	-	-	-	-	-	-	-	-

## Data Availability

Data are contained within the article and Appendix A.

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
