# Peer review of "Propolis Alleviates Acute Lung Injury Induced by Heat-Inactivated Methicillin-Resistant Staphylococcus aureus via Regulating Inflammatory Mediators, Gut Microbiota and Serum Metabolites"

_nutrients, 2024, doi:10.3390/nu16111598_

Round 1

Reviewer 1 Report

Comments and Suggestions for Authors

It is an interesting paper with some minor concern

Fig 1 will be improved in my opinion if you show the mouse groups in the figure.

Was a sample size calculated?

After ANOVA how was the post hoc 2x2 test performed? Was a correction applyed to reduce alpha error inflation?

Comments on the Quality of English Language

the language is clear in my opinion

Author Response

1: Fig. 1 will be improved in my opinion if you show the mouse groups in the figure.

Response 1: We appreciate the reviewer’s comments. Following the reviewer’s comments, we have revised Figure 1, which could be found in line 144, page 4.

2: Was a sample size calculated?

Response 2: We appreciate the reviewer’s comments. In conjunction with previous studies and experimental expectations, we calculated the required sample size using the G*Power software with the following parameters: a significance level of 0.05, a statistical power of 0.80, and an estimated effect size of 0.5. The calculations showed that to detect a significant effect with enough power, we would need to have at least 9 mice in each group when using one-way analysis of variance (ANOVA).

3: After ANOVA how was the post hoc 2x2 test performed? Was a correction applyed to reduce alpha error inflation?

Response 3: After ANOVA and finding significant overall differences between groups, we performed post hoc tests using Tukey's Honestly Significant Difference. If the data were not normally distributed, we used the Kruskal-Wallis test. We have revised in line 217-220 of the manuscript.

Reviewer 2 Report

Comments and Suggestions for Authors

Li et al.'s manuscript analyses propolis's effect on MRSA-induced lung injury. The analyses focused on proinflammatory mediators' response, gut microbiota, and serum metabolites. The manuscript is interesting; the effect is analyzed from different perspectives. However, some questions need further clarification.

1. Were the mice exposed to antibiotics before treatment? If not, why would the host activate a different response to MRSA compared to other strains of S. aureus? Please give more justification at the end of the Introduction for strictly testing this species.

2. L28-L32: Please, give more citations

3. Please add a source of propolis to the research papers mentioned in the Introduction.

4. The ethanol extract of propolis was used in the experiment. How can you exclude the effect of ethanol?

5. The paper's title talks only about propolis, but in the abstract, it also appears to be tree gum. What is the connection between these two components?

6. L61: is it more effective?

Author Response

RE: Manuscript Number: Nutrients-2997097

MS Title: Propolis alleviates acute lung injury induced by heat-inactivated methicillin-resistant Staphylococcus aureus via regulating inflammatory mediators, gut microbiota and serum metabolites

Zongze Li, Zhengxin Liu, Yuyang Guo, Shuangshuang Gao, Yujing Tang, Ting Li and Hongzhuan Xuan *

Nutrients

Dear editor,

Enclosed is our revised manuscript entitled “Propolis alleviates acute lung injury induced by heat-inactivated methicillin-resistant Staphylococcus aureus via regulating inflammatory mediators, gut microbiota and serum metabolites” (Nutrients-2997097) by Zongze Li, Zhengxin Liu, Yuyang Guo, Shuangshuang Gao, Yujing Tang, Ting Li and Hongzhuan Xuan*. We hope this revision will meet your final approval for Publication in Nutrients. According to the comments made by the editor and reviewers, we have made some changes in the revised manuscript as follows. And all the changes in the manuscript have been highlighted. I confirm that all authors have agreed to authorship and order of authorship for this manuscript and that all authors have the appropriate permissions and rights to the reported data.

1: Were the mice exposed to antibiotics before treatment? If not, why would the host activate a different response to MRSA compared to other strains of S. aureus? Please give more justification at the end of the Introduction for strictly testing this species.

Response 1: We appreciate the reviewer’s comments. MRSA is resistant to conventional antibiotics, making its infections more difficult to treat compared to other Staphylococcus aureus infections. Clinically, both MRSA and Staphylococcus aureus can cause lung infections and an increase in inflammatory factors including TNF-α, IL-1β, and IL-6. Moreover, MRSA can lead to a decrease in unclassified_f_Lachnospiraceae in the gut microbiota, as well as a reduction in butyric acid in short-chain fatty acids. In our study, we observed varying degrees of lung damage in mice following injection with heat-inactivated MRSA. Levels of inflammatory factors TNF-α, IL-1β, and IL-6 were elevated. Importantly, our analysis of gut microbiota also revealed a decrease in the abundance of unclassified_f_Lachnospiraceae and a reduction in butyric acid in short-chain fatty acids. Following the reviewer’s comments, we have revised the manuscript in line 77-85.

2: L28-L32: Please, give more citations.

Response 2: We appreciate the reviewer’s comments. Following the reviewer’s comments, we have added citations in lines 28-35.

3:Please add a source of propolis to the research papers mentioned in the Introduction.

Response 3: We appreciate the reviewer’s comments. Following the reviewer’s comments, we have revised the manuscript in lines 72-76.

4: The ethanol extract of propolis was used in the experiment. How can you exclude the effect of ethanol?

Response 4: We appreciate the reviewer’s comments. Prior to gavage, EEP and EEG were dissolved in gum tragacanth, respectively, and the final concentration of ethanol in the gum tragacanth was less than 1% (v/v). We have revised the manuscript in line 108-110.

5: The paper's title talks only about propolis, but in the abstract, it also appears to be tree gum. What is the connection between these two components?

Response 5: We appreciate the reviewer’s comments. Propolis is a resinous substance collected by Apis mellifera from various tree gum for use as a sealant in the hive. We analyzed the differential components of propolis and tree gum in line 226-235. A total of 1635 metabolites were detected, with 1040 being differential metabolites, primarily flavonoids and phenolic acids. The antibacterial mechanisms of flavonoids such as robinetin, myricetin, apigenin, rutin, and galangin had been previously reported. Phenolic acids, including chlorogenic acid, protocatechuic acid, p-coumaric acid, caffeic acid, ferulic acid, and gallic acid, had also been documented for their ability to inhibit bacterial growth and reproduction. In this study, these differential metabolites were found to have relatively higher relative contents in propolis. Analyzing the chemical components helps us distinguish between propolis and tree gum.

Comments 6: L61: is it more effective?

Response 6: We appreciate the reviewer’s comments. We have revised the manuscript in line 61-62.

Reviewer 3 Report

Comments and Suggestions for Authors

The work submitted for review has been very carefully planned and prepared. The introduction has been well prepared. Methods adequately described. The results are supported by many graphs and fairly well discussed. The work is generally publishable in this form. However, I have a request to the authors to respond to a few minor points:

1.Did the authors evaluate the cytotoxicity of EEP and EEG? Up to what concentrations are they non-toxic?

2. Did the authors assessed the antibacterial activity of EEP and EEG against the MRSA strain used (MIC and MBC)?

3. I am missing an explanation of what specific compounds present in EEP and EEG determine their stronger/weaker biological activity?

Author Response

RE: Manuscript Number: Nutrients-2997097

MS Title: Propolis alleviates acute lung injury induced by heat-inactivated methicillin-resistant Staphylococcus aureus via regulating inflammatory mediators, gut microbiota and serum metabolites

Zongze Li, Zhengxin Liu, Yuyang Guo, Shuangshuang Gao, Yujing Tang, Ting Li and Hongzhuan Xuan *

Nutrients

Dear editor,

Enclosed is our revised manuscript entitled “Propolis alleviates acute lung injury induced by heat-inactivated methicillin-resistant Staphylococcus aureus via regulating inflammatory mediators, gut microbiota and serum metabolites” (Nutrients-2997097) by Zongze Li, Zhengxin Liu, Yuyang Guo, Shuangshuang Gao, Yujing Tang, Ting Li and Hongzhuan Xuan*. We hope this revision will meet your final approval for Publication in Nutrients. According to the comments made by the editor and reviewers, we have made some changes in the revised manuscript as follows. And all the changes in the manuscript have been highlighted. I confirm that all authors have agreed to authorship and order of authorship for this manuscript and that all authors have the appropriate permissions and rights to the reported data.

1: Did the authors evaluate the cytotoxicity of EEP and EEG? Up to what concentrations are they non-toxic?

Response 1: We appreciate the reviewer’s comments. We had evaluated the cytotoxicity of poplar propolis in previous studies. In our previous article titled "Bioactive components and mechanisms of poplar propolis in inhibiting proliferation of human hepatocellular carcinoma HepG2 cells (Biomedicine & Pharmacotherapy, 2021, 144, 112364)," poplar propolis showed little cytotoxicity to LO-2 cells at the concentration of 50 μg/mL. Similarly, in another article titled "Antitumor Activity of Chinese propolis in human breast cancer MCF-7 and MDA-MB-231 cells (Evidence-Based Complementary and Alternative Medicine, 2014, 2014, 280120.)," poplar propolis also exhibited little cytotoxicity to HUVEC cells at the concentration of 50 μg/mL. We extrapolated the dosage for mice based on this concentration. Additionally, our study referenced other animal experiments related to poplar propolis. The references were cited in lines 140-141 of the manuscript.

2: Did the authors assessed the antibacterial activity of EEP and EEG against the MRSA strain used (MIC and MBC)?

Response 2: We appreciate the reviewer’s comments. In our previous article titled " Antibacterial activity of Chinese propolis and its synergy with β‑lactams against methicillin‑resistant Staphylococcus aureus (Brazilian Journal of Microbiology,2022, 53(4):1789-1797)," we have assessed the antibacterial activity of EEP. In another article we are currently publishing, we conducted in vitro antibacterial experiments with EEP and EEG against MRSA. The results showed that the MIC of EEP was 0.15625 mg/mL and the MIC of EEG was 0.3125 mg/mL. The MBC of EEP was 0.3125 mg/ml, while the MBC of EEG was 0.625 mg/ml.

3: I am missing an explanation of what specific compounds present in EEP and EEG determine their stronger/weaker biological activity?

Response 3: We appreciate the reviewer’s comments. The components of EEP and EEG are highly complex. The biological activity of propolis is determined by a variety of constituents. We identified their chemical compositions through UPLC-ESI-QTRAP-MS/MS, with the main differential metabolites being flavonoids and phenolic acids (line 225-234). The antibacterial mechanisms of flavonoids such as robinetin, myricetin, apigenin, rutin, and galangin had been previously reported. Phenolic acids, including chlorogenic acid, protocatechuic acid, p-coumaric acid, caffeic acid, ferulic acid, and gallic acid, had also been documented for their ability to inhibit bacterial growth and reproduction. In this study, these differential metabolites were found to have relatively higher relative contents in propolis. Therefore, the quantity and relative content of flavonoids and phenolic acids may determine the differences in their biological activities.